# Competition Prospects in Regulated Market of a Baltic Country

Aurelija Burinskienė

Department of Business Technologies and Entrepreneurship, Faculty of Business Management, Gediminas Technical University, Saulėtekio Av. 11, LT-10223 Vilnius, Lithuania; aurelija.burinskiene@vilniustech.lt

**Abstract:** The author examines competition prospects in the regulated market of a selected EU country to investigate specific policy choices. Under full market regulation, the actors have the exclusive right to deliver economic transactions. (1) Literature review: The theoretical part starts with the presentation of actors' heterogeneity and their market competition. (2) Methods: During empirical research, the author revised the cost structure and tested assumptions of the new trade theory of heterogenous firms and the impact of this heterogeneity on actors' competition. The author formed a three-step methodology where at the beginning, the actors are described, then a regression model is formulated, and volumes of delivered services are analyzed. In the end of this study, the author investigated if the costs of regulated market participants are characterized by the distribution of Pareto and analyzed the gaps between regulated prices and the level of costs. The author applied Pareto distribution and LS regression methods in the study. (3) Results: Although the author found that the market participants in the market concerned are very different, their distribution according to the Pareto distribution function could not be confirmed. (4) Conclusions: The study shows that competition prospects in a regulated market differ. This is proven by analyzing the impact of this heterogeneity on actors' competition. These results are essential in formulating the suggestion of economic policy.

**Keywords:** competition prospects; regulated market; regulation; Baltic country

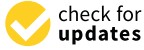



## 1. Introduction

Krugman (1979, 1980) formalized that economies of scale and imperfect competition can stimulate activity without comparative advantage for the parties. After Krugman published his pioneering work, the new trade theory quickly spread. Based on this theory, Krugman and other like-minded people developed and further developed models that define the market of monopolistic competition.

A country appoints actors, thereby regulating their number and granting them exile rights to produce authentic deeds. In a regulated system, high-level professional actors provide services to produce original acts and certify various activities, which are important economic transactions.

As in other countries belonging to this system, the decisions of actors having exclusive rights in the selected country have the same legal force as court decisions. Competition between actors is largely eliminated: the activities of actors are limited to the territory assigned to them, and fixed rates or rates that may change only within certain price limits shall be set (Spence 1976).

However, the changes due to Industry 4.0 technologies are inevitably already in place and will affect each sector in the future (Holmes 2011; Kline et al. 2021). Already, the most prominent actors stand out, who are distinguished by vast volumes of contracts and demonstrate high productivity (Foster et al. 2008; Fox and Smeets 2011). Moreover, how the actors would change if territorial restrictions ceased to apply in the virtual space to provide regulated services here is already being considered. Hopefully, this could improve the economic situation of actors operating in far-flung and less economically active regions and attract those who want to work there and provide regulated services. This is very

important for the state seeking to ensure the availability of regulated services in remote areas (Pagliero 2011).

In this study, the author aims to assess competition in the state-controlled market for regulated services, including the possible changes mentioned above, which may be necessary for the gradual liberalization of this market. This empirical study applies a competition model between companies based on the new trade theory. At the same time, the author examines how the different skills demonstrated by actors in the market affect the value of their business.

This research aims to answer these research questions:

- Is it possible to identify the gaps between regulated prices and the level of costs?
- Are the costs of regulated market participants characterized by the distribution of Pareto?

To answer these questions, the author applies Pareto distribution and LS regression methods in this study.

The paper consists of six sections. The introduction focuses on actuality and introduces the paper's research area. Section 2 is dedicated to the literature review. An overview of competition prospects is provided here. The outline and proposed methodology that can be useful for the revision of the competition is presented in Section 3. Section 4 describes the results of empirical research. Section 5 is given to the discussion. The study's limitations and future plans are presented in the discussions. Finally, the paper ends with conclusions, which are provided in Section 6. The summary and limitations of the research are presented in the last section.

Increasingly concerned with dynamic efficiency, the institutions guaranteed for an economic policy like to see whether concrete policy decisions can lead to greater prosperity in the long run. It must be evaluated how and if specific actions can develop a market structure to achieve greater welfare. The development of the market structure is resulting in a change in the competition between actors. This paper strives to give an investigation important for economic policy formulation.

In the next section, the author will provide the links between the heterogeneity of actors and market liberalization, which has an impact on actors' competition.

## 2. Literature Review

The literature review provides insights into heterogenous actors' competition and how regulation of the market impacts their productivity, and presents the experience of the liberalization of existing regulations in several European Union countries (mainly in France and shortly in Portugal, Poland, Austria, Estonia, the Netherlands and Italy).

A large body of well-founded empirical research by renowned economists confirms that firms are highly heterogeneous across many parameters of their performance. Therefore, when such companies are examined, they are often called heterogeneous.

Companies can differ according to many operational parameters: the company's status as per revenue, productivity, the level of capital and employees' skills and the efficiency of their use, the size of the firm (both in terms of assets and in terms of income), product quality and uniqueness from the point of view of consumers, etc. Data on the heterogeneity of firms began to be collected more intensively in the 1980s (Redding 2010).

Since the last decade of the last century, empirical studies have been carried out on the heterogeneity of companies in terms of their productivity. The aim was to find out the correlations between productivity and revenue.

The heterogeneity of firms, especially in terms of productivity, is also essential when the market is liberalized: redistribution occurs not so much between industries, but within them, with the exit of less productive actors from the market (this is shown by empirical data in developing economies (Tybout and Westbrook 1991; Pavcnik 2002), as well as in countries considered to be developed (Bernard et al. 2006). As observed by Melitz (2008), market liberalization creates opportunities for more productive firms to gain more market share at the expense of less productive firms.

Helpman et al. (2003), in their review of empirical studies, pointed out that the productivity distribution of firms in an industry in the medium term follows and can be mathematically described using a Pareto distribution function. In other words, there are more companies with lower productivity in the market, which, unable to cover the fixed costs of exports and direct investments abroad, focus only on the country's domestic demand.

For several decades, economists in industrial organizations have tried to formulate suitable criteria by which to judge the effectiveness of competition. This has resulted in an extensive list of market structures and behavioral endeavors that are expected to be associated with the effective competition. The resulting structure–behavior–performance paradigm provides a framework (rather than an exact criterion) for evaluating competitive prospects.

It should be noted that the recent reforms of regulated service fees in various states are based on the desire that the level of fees should be related to the cost of performing transactions. Such an aspiration manifests itself in two ways: (1) proponents of the functioning of competition law in the regulated market advocate that fees should not be regulated at all or regulated as little as possible (for example, without a minimum rate, thus leaving the possibility of lowering prices) and that the price of transactions should be determined by the market (De Jan et al. 2016); (2) regardless of how the status of actors is assessed from the point of view of competition law, it is argued that states must regulate the level of fees, but this must be determined by taking into account economic data on the cost of performing transactions (Dixit and Stiglitz 1977).

The view that, under market conditions, the issue of fees should not be regulated by the state is reflected in the reforms of fees carried out in the Netherlands and Italy (the level of fees is no longer fixed). The conditions for the number of specific rates to be determined by the functioning of the market are also created by those countries that provide only the maximum number of rates (for example, Poland) or only the recommended level of fees (for example, in Austria, the rates enshrined in the Law on regulated Tariffs are the maximum, but not mandatory rates for actors to follow) (Mankiw and Whinston 1986).

The view that decisions on the regulation of fees must be based on data on the costs incurred by actors is reflected in the reform of fees in France. With this approach, the plausible conclusion is the recognition of the possibility and/or need to reduce some fees, since research data show that the fees for actors' actions for the formalization of individual types of transactions (in the EU countries examined by the researcher) are much higher than the cost of performing their acts.

The researchers' assessments of whether the manifestations of the emphasis on the cost criterion in states' fee reforms can be seen as a trend toward the liberalization or deregulation of the regulated market in the European Union are different (Best and Kleven 2018).

In cases where reforms to the liberalization of the regulated market are carried out, they are characterized by a complex nature, i.e., the issue of prices is one of several issues to be resolved (others are the question of the placing in the market of new actors, and the refusal of the binding nature of some transactions).

On the other hand, there is also the opposite trend in Europe, namely, to ease the burden on courts and governments by transferring additional functions to actors (e.g., issuing a European certificate of succession and formalizing agreement between the parties).

It is typical for EU countries that the competition authorities of those countries promote the liberalization of regulated markets (e.g., Estonia). On the other hand, the example of France shows that the involvement of the competent body in the reform of fees helps to achieve not so much the liberalization of rates as the correct (not too small and not too high) determination of the level of fees (Bresnahan and Reiss 1991).

An important aspect of the recent reforms in fees is the desire to ensure universal access to regulated services, both in terms of the level of fees and in terms of geographical accessibility (e.g., Portugal, France). One of the issues addressed during the reforms is the

question of the different profitability of regulated activities in different regions of the states. It is dealt with in different ways: by facilitating the possibility of setting up actors in less profitable regions (France), by applying different rates (France: separate rules for overseas territories), and by using models of solidarity between actors, in which compensation is paid from the activities of actors in more prosperous regions through self-governing bodies to actors working in less profitable regions (e.g., Portugal).

The reform of fees in France was carried out in 2015 with the adoption of the Law on the Equality of Economic Growth, Activity, and Economic Opportunities. A separate section of the law defines the conditions of activity of regulated legal professions (actors). As regards the reform of the activities of actors, the law aimed to improve the geographical accessibility of transactions (promoting the establishment of new actors' offices in less prosperous regions to facilitate the possibilities for young people, especially women, to become actors), while maintaining the existing economic viability of actors' offices, especially in rural areas. The reform of fees was not an independent goal of the reform, but the more apparent linking of fees to costs was perceived as a means of achieving the other objectives of the reform and, with it, economic regulation.

Following the adoption of the Macron Law, actors' fees must be determined considering the related costs of the service provided and reasonable remuneration. The fee shall be determined based on objective criteria based on the available data that the national professional authorities are obliged to provide. This pricing method also ensures cross-subsidization so that transactions are accessible to all (Bresnahan 1982).

In 2016, as part of implementing the Regulation Act in France, the Government decided to apply the method of determining the rates for each individual transaction (act) according to the costs incurred. This method turned out to be inappropriate since no such cost accounting system would allow such an accurate assessment of the associated costs. As a result, the Law of 23 March 2019 changed this cost calculation method. Both the cost and the reasonable salary are calculated for the actors of employees. The French competition authority observed that the charges could be set in one of those, acting based on costs, or measuring the duration of each act, or calculating the costs incurred. It is recommended the second option, precisely since collecting the necessary data in the first version would be difficult. The competition body also believed that fixing a tariff based on average costs would maintain (as opposed to the determination of the rate based on actual costs) the incentive for an individual professional to reduce their costs and, thus, generate higher profits.

The amount of reasonable remuneration is defined in the Code of Commerce as 20% of the profit, multiplied by a coefficient from 1 to 1.6 (determined according to the indicators defined in the Code of Commerce when evaluating a period of three years) and applying other conditions established by the Code of Commerce. In 2020, the amount of reasonable remuneration calculated for actors (defined in the by-law) was 30%, and in 2022 it was 32.7%. It is noteworthy that, in conclusion, the French competition authority explained that the fees charged for the different treatments must be set in such a way as to enable the professionals to cover their operating costs and obtain a reasonable profit (margin). This profit must compensate the professional for invested capital and the work they do. Therefore, when setting tariffs based on costs (costs) and reasonable profits, fair remuneration for capital and reasonable compensation for professional activity (work) must be established.

### 3. Related Literature and Outline

There are several literary directions. First, much literature shows that free market entry can be too high when companies have market power. Literature on limited market access is more limited; see Sejm and Waldfogel's (2013) work on analyzing the state monopoly. For comparison, the author examines the private monopoly, when the state issues new entry licenses every year and in coordination with the sector in different places. Thus, the

analysis presented in this paper can show to what extent the state is regulated, as in the hypothesis of Stigler (1971).

Secondly, the literature on professional licensing examines the effects of general requirements (Schmidt et al. 2007; Kleiner and Soltas 2019), but the market structure is not profoundly revised.

Thirdly, the literature was revised from a methodological point of view. This literature focused on setting fixed cost limits using immediate inaccuracies in the conditions for profit maximization, as in Pakes (2010) and Pakes et al. (2015). For the latest programs, see, for example, Eizenberg (2014), Berry et al. (2016), Fan and Yang (2020), Ishii (2005) and Wollmann (2018). Compared to their work, this research aims to use the initial model to derive both fixed and variable costs.

After reviewing the literature dedicated to the study of the productivity of actors, it can be observed that the models describing the heterogeneity of companies according to productivity parameters do not always correspond to the results of the empirical studies on the heterogeneity of companies. It is not uncommon to even come to opposite conclusions, contradicting the evidence provided by well-founded empirical studies.

Empirical studies, which examined the heterogeneity of companies from various aspects, found that the competitive pressure experienced by the company from its competitors, resulting from the restrictions placed on the activities of competitors, forced the company to innovate to achieve higher productivity, and only because of this did the competitive advantage allow the company to increase its market power.

Empirical studies have established that the distribution of companies according to productivity in industries often corresponds to the Pareto distribution function. According to this distribution, the most productive firms represent only a small fraction of the entire industry, while the less productive firms constitute a significant majority. Thus, it is more likely that precisely the most productive firms expand and consolidate their competitive advantage. The company's choice should also depend on its available production capacities, the amount of investment in them and the planned profit.

It should be noted here that the heterogeneity of companies has been modeled for twelve years in the models of heterogeneous companies of the new trade theory, which, as well as in the model of market equilibrium, is modeled under the conditions of oligopolistic competition, or more precisely, monopolistic competition. Therefore, it can be assumed that the models of definitely heterogeneous companies could be used by adapting them for competition policy.

This study aims to determine the practical actions necessary for decision-making helping to achieve competition in the market.

The revision is settled in three steps:

- Revision of the costs structure of the actors;
- Identification of links between variables and construction of regression model;
- Revision of the effect of volume-oriented service.

The author divided the methodology into three steps representing the research design (see Table 1).

Table 1 provides the connection between competition in the market and economic policy, with the help of a three-step methodology, providing descriptions, relationships and methods specific to each step.

For the third step, the author formulated several research questions that the author aims to answer during the study:

- Is it possible to identify the gaps between regulated prices and the level of costs?
- Are the costs of regulated market participants characterized by the distribution of Pareto?

In the next section, the author provides the results of empirical research.

**Table 1.** Methodology highlighting the connection between competition in the market and economic policy.

| Steps | Objectives | Description | The Application of Methods | Links with Economic Policy |
|---|---|---|---|---|
| 1st step Revision of actors' costs structure | Revision of productivity level in the regulated market. | • Distribution of costs. | • Review of best practices stated in literature; • Descriptive statistics. | • The decision helps to increase competition. |
| 2nd step Formulation of regression model | Identification of links between variables. | • Characteristics of actors. | • Panel data analysis; • LS regression method. | • Investigation of regulation effect on the market. |
| 3rd step Analysis of services and characteristics of the market | Revision of service delivery volumes. | • Impact of volumes. | • Comparison using simple moving average function; • Investigations; • Pareto distribution. | • Determining volume-related aspects. |

## 4. Results

For this study, the author does not model the demand that significantly exceeds the supply in the regulated market and, therefore, forces the government to consider a possible increase in the number of firms and follow the economic policy formulation accordingly. Another reason for not modeling demand is that the state sets the price in a regulated market. Therefore, it cannot be influenced by consumers and providers of regulated services (Farronato et al. 2020).

As mentioned, the analysis in this paper focuses on the state-regulated sector of the selected country, where the government sets prices, the number of firms and the territory of their activities.

The database used for research consists of the financial data of actors' income declarations on their expenses and the data available on the number of transactions performed and the income received by individual groups of services. The collected data cover one year period, and such a data set is appropriate for the calculation of Pareto distribution. The number of observations was 487.

The author uses these data for research purposes, in contrast to the selected country's legislation of regulated services. After that, for each of these groups, the author calculates the variable and fixed costs in the order described here in detail. The author grouped all the lines of actors' costs according to their relationship with the number of transactions: the author classified the charges that showed a statistically significant association as variable, and others as fixed costs.

Using these datasets, the author applied such methods as descriptive statistics (whose results will be presented in Tables 2 and 3), panel data analysis (whose results are used to construct a regression model), Pareto and LS regression (whose results will be presented in Equation (1)). The research results are given below and start with the presentation of descriptive statistics (as mentioned in the first step of the methodology).

**Table 2.** Variable and fixed cost regression against the total number of regulated services.

| Variables against the Total Number of Regulated Services | Coeff. | t-Statistic | Adj. R$^2$ | Centered VIF | Breush-Godfrey LM Test | | ARCH | |
|---|---|---|---|---|---|---|---|---|
| | | | | | F-Statistic | Prob. | F-Statistic | Prob. |
| Variable costs | 0.924948 | 20.82128 | 0.5964 | 1.0373733 | 0.16305 | 0.6865 | 0.261493 | 0.6093 |
| Fixed costs | 0.026509 | 0.424458 | | | | | | |

Source: calculated by the author.

**Table 3.** Descriptive statistics.

| Variables | Mean | Median | Max | Min | Std. Dev. | Skewness | Kurtosis | Jarque-Bera | Prob. |
|---|---|---|---|---|---|---|---|---|---|
| Revenue of regulated services | 70,818 | 61,658 | 436.154 | 5077 | 45,412 | 2.94 | 17.72 | 5095 | 0 |
| Variable costs | 54,769 | 39,768 | 404.800 | 0 | 48,077 | 2.46 | 12.18 | 2199 | 0 |
| Fixed costs | 25,040 | 19,137 | 187,902 | 0 | 22,921 | 3.17 | 18.54 | 5719 | 0 |
| Total number of regulated services | 6326 | 5795 | 17,719 | 497 | 2907 | 0.91 | 3.91 | 84 | 0 |

Source: calculated by the author.

Later, during the second step of the methodology and following panel data analysis, the author used the LS regression method and modeled the profit of each office *i* as the difference between variable profits $\pi_i^v$ and fixed costs $f_i$:

$$\pi_i = \pi_i^v - f_i = p^1 q_i^1 - w l_i \left( q_i^1, q_i^2 \right) - m_i \left( q_i^1, q_i^2 \right) - f_i \tag{1}$$

where $p^1 q_i^1$ is for revenue from regulated services; $w$—wages paid for employees; $l_i$—labor force employed; $m_i$—goods and services provided by offices.

To evaluate the amount of labor force, the author classified total payrolls (taxes included) against volumes of regulated services. This gives us a rough estimation of the number of employees and their working hours (see Table 4).

**Table 4.** Labor force estimation.

| Q | Mean | Std. Dev. | Obs. | Payrolls per Month, Eur | | |
|---|---|---|---|---|---|---|
| | | | | Per firm | No of Employees | Wage per Employee |
| [0, 2000) | 10,652 | 10,589 | 12 | 888 | 1 | 888 |
| [2000, 4000) | 12,968 | 7737 | 92 | 1081 | 1 | 1081 |
| [4000, 6000) | 21,542 | 13,866 | 154 | 1795 | 2 | 898 |
| [6000, 8000) | 33,289 | 18,545 | 111 | 2774 | 3 | 925 |
| [8000, 10,000) | 44,859 | 25,142 | 65 | 3738 | 3 | 1246 |
| [10,000, 12,000) | 74,083 | 44,940 | 26 | 6174 | 4 | 1543 |
| [12,000, 14,000) | 93,058 | 35,673 | 16 | 7755 | 5 | 1551 |
| [14,000, 16,000) | 102,918 | 41,452 | 8 | 8577 | 5 | 1715 |
| [16,000, 18,000) | 127,276 | 49,214 | 3 | 10,606 | | 1231 |
| All | 32,586 | 30,103 | 487 | | | |

Source: calculated by the author.

Having collected all the data, the author calculated average prices and variable and fixed costs. In Table 5, the author provides variable cost distribution against fees. These data provide a clear picture of how margins are distributed in the regulated system.

**Table 5.** Variable costs against prices. Price levels of regulated services, EUR.

| Costs, EUR | Mean | Std. Dev. | Obs. |
|---|---|---|---|
| [0, 50] | 24.98 | 9.08 | 447 |
| [50, 100] | 62.11 | 13.00 | 39 |
| [100, 150] | 113.74 | NA | 1 |
| All | 28.14 | 14.34 | 487 |

Source: calculated by the author.

For the third step of the methodology, smoothed[1] series of prices and total costs against the number of provided services are provided in Figure 1. Since fixed percentage rates are established, the prices for the approval of transactions grow with an increase in the number of transactions. This shows that offices that provide regulated services in regions with increased activity (primarily in major cities) benefit more from the increased volume of regulated services.

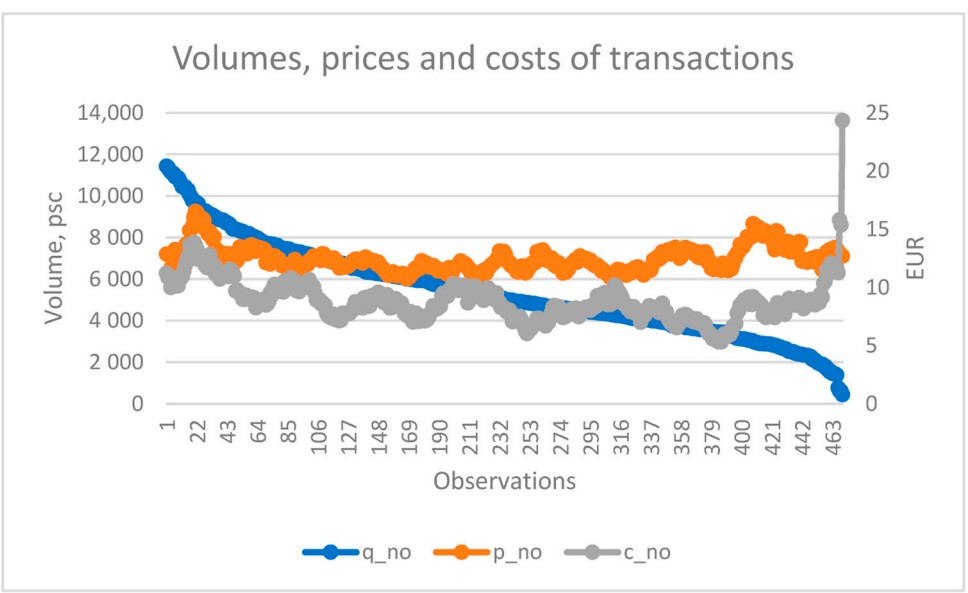

**Figure 1.** Price and cost levels against the numbers of regulated services transactions. Source: calculated by the author (where q—quantity, p—price, c—cost).

However, the exact opposite is true with other regulated services, where the average gap between prices and the level of costs is relatively small. However, it should be noted that actors with smaller amounts of regulated services apply higher margins than their counterparts who provide more services.

In this study, in which the author examined the supply side of the regulated market, the author found no confirmation that, according to the new trade theory of heterogenous firms, the costs of market participants are characterized by the distribution of Pareto (see Figure 2).

Meanwhile, in the case of the regulated market in question, especially in the area of confirming transactions, only a few offices of the regulated market are distinguished by meager costs (two to three times lower than the average).

According to economic theory, most market participants are distinguished by high variable costs (low productivity). In contrast, larger companies, as a rule, are characterized by exceptional productivity. There are few market participants, but a significant competitive advantage distinguishes them. These insights are essential for economic policy formulation.

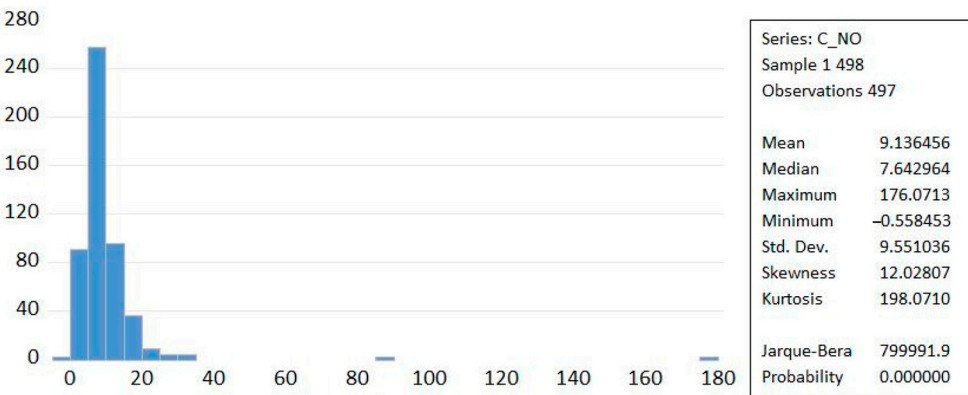

**Figure 2.** Price and cost levels characterized by the distribution of Pareto.

## 5. Discussion

The market liberalization, which promotes competition between the firms, should benefit customers, because the quality of services is improved, and services costs are reduced due to its liberalization Kim (2006). However, a market liberalized too quickly, where risk assessment is left to the market alone, can lead to boom-and-bust cycles caused by instability, which can have a negative impact on the market (Semmler and Young 2010).

Other authors mention the fact that liberalization policies have encouraged local governments to improve enterprise performance, which is directly related to their own revenues. However, local government incentives could lead to various institutional arrangements designed to reduce transaction costs and agency problems, which also create highly competitive markets and strong incentives for firms Park et al. (2006).

The authors of (Lall and Latsch 1998) suggest thinking about controlled liberalization, recommending the growth in labor productivity and resource-based activity, with some input to more complex activities, which help firms to become competitive in international markets.

In this research, the author applied the theoretical models of new trade and the empirical evidence of the heterogeneity of companies in the markets. Based on these fundamental and subject studies, the author proposes an applied economic policy model for a regulated market, considering the existing distribution of enterprises in that market in terms of efficiency and possible redistribution after market liberalization. This modelling approach is different from the modelling approach of the authors of Costantini and Melitz (2008) where they also include the actors' costs which are necessary for entry to the market. By using the proposed economic policy model, the author examines how this could affect the value of companies in the transition from a regulated to a free market. Following the investigations of other authors Lall and Latsch (1998), it is necessary to design how actors can enhance their technological and service internationalization drive, and what the role of the government is in facilitating this drive.

The author has noticed that actors with the highest volumes of regulated services are distinguished by extremely high productivity, which, the author believes, clearly indicates the application of more advanced technologies (the author thinks these may be the first manifestations of Industry 4.0 in this market). Therefore, with the liberalization of the need for regulated services, such firms would have a significant competitive advantage, which could be realized for the concentration of market power. The research results will help prepare for the regulated market's liberalization. A further research direction could include modeling the demand for transactions that are liberalized and have no regulated fees.

As the market still has high variable costs (which shows a low productivity case), before liberalization, it is recommended for government to take the controlled liberalization approach (previously suggested by Lall and Latsch (1998)) or the partly regulated market approach (previously suggested by Hyman (2000)), which could allow the slight adaptation of the actors to the changes of the market.

## 6. Concluding Remarks

The organization of industry and the new theories of trade offer many incentives, such as competition and market structure changes, including investment and innovation activities.

While the discussion continues, theoretical and methodological studies focus on an integrated approach. Without detailed theoretical models, it is difficult for the researcher to give some evidence on the topic and recommendations for economic policy formulation.

To properly define econometric structural models of the market balance, imperfect competition between firms must be considered. Therefore, data modeling methods are often used.

The construction of the methodology allows for easily verifiable results, which can be important when testing the proposed empirical research design in practice. By applying analysis, the author sees that the costs do not meet the Pareto distribution as suggested in the theoretical model of Melitz.

The author proposes a staged methodology, which helps to highlight the connection between competition in the market and economic policy.

The methodology is composed of the following stages: revision of the costs structure of the actors; identification of links between variables and construction of regression model; revision of the effect of volume-oriented service.

The application of the methodology is evident in this paper, where the author presents the results of empirical research. The proposed methodology could be applied further for the analysis of the structure of regulated markets. Until now, only methodology for trade liberalization evaluation was suggested by Estrada (2004), not trade liberalization preparation. This is particularly important when we see the reducing number of regulated markets in the European Union (World Bank Group 2020).

For further steps, it is recommended to implement a partly regulated economic policy, which would support slight market evolution from fully regulated to non-regulated.

The limitation of this research is that this study does not focus on demand, because in the regulated market, the price level is fixed, i.e., demand does not depend on the price level.

**Funding:** Not applicable.

**Institutional Review Board Statement:** Not applicable.

**Informed Consent Statement:** Not applicable.

**Data Availability Statement:** Not applicable.

**Conflicts of Interest:** The author declares no conflict of interest.

## Notes

[1] With simple moving average function.

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
