# Peer review of "Competition Prospects in Regulated Market of a Baltic Country"

_economies, doi:10.3390/economies11020052_

Round 1

Reviewer 1 Report

COMMENTS FOR THE AUTHOR:

Competition Prospects in Regulated Latin Notary Market of Baltic Country

Broad comments

The manuscript contributes an interesting study on competition prospects in the state-controlled market for notarial services assessing the possibility of the gradual liberalization of the market. Unfortunately, it does not follow scientific writing guidelines. The manuscript should be substantially improved to be eligible for publishing.

General

The manuscript lacs a read thread of reasoning. The sections are set independently and do not form coherent reporting.

Recommendation: Adding research questions or/and hypotheses in the research design and presenting them in the introduction; beginning sections with an introductory paragraph and finishing them with a paragraph of synthesis and assuring logical connection between paragraphs.

Abstract

The abstract merely describes the research steps without presenting meaningful results and contributions.

Recommendation: Rewriting the abstract, considering the research question and/or hypotheses.

Introduction

The introduction provides only information on the practical background and the aim of the research.

Recommendation:

The introduction should provide information on theoretical and practical background with:

-        general reserach framework, backgrund,

-        general research challenge, problem,

-        the aim or objective of the research,

-        the hypotheses or/and research questions,

-        list of methods (data provision, analysis).

Literature review

The literature review provides comprehensive information on the notarial market issues in different countries. However, it lacks explaining the meaning of the discussion for the research. As the rest of the manuscript does not refer to it, it seems the content of the second section is more or less insignificant for the research.

Recommendation: Try to refer to the second section in the discussion.

Related literature and outline

The literature is merely a listing of previous research.

Recommendation: The literature research should report on objectives, hypotheses or/and research questions, theoretical ground, conceptual framework, and operationalization. The section should end with an explanation of the research design justified by previous research (suitable articles listed in the section)

Empirical dataset and framework

The section explains the research steps without a methodological framework.

Recommendation: At the beginning, add a short description of the methodological framework in connection to research questions and/or hypotheses. Structure the report along with the research question or/and hypotheses.

Discussion

The discussion is short and narrow. Lacs of explaining the contribution (practical and scientific) Lacks referencing other research results.

Recommendations:

Recommendations: A scientific conclusion has to reference other research results and place the presented study into a specific research environment.

Author Response

Thank you for valuable comments

Reviewer 2 Report

The study fails to provide a rationale for studying this topic and its novelty. 

The introduction should detail the research question, motivation and contributions. It is not clear which Baltic country the study is about? The introduction needs more work. 

No mention of the source of data and country in Section 4 is very confusing. Need a detailed discussion of source and country. Also the rationale for picking the data source. 

There should be seperate sections on Results, Discussion and Conclusion. The current form is quite weak and needs much more rigorous details. 

Author Response

Thank you for valuable comments

Reviewer 3 Report

The paper discusses competition prospects in regulated Latin notary market of selected Baltic country. The work is very interesting and has traits of originality.
The literature review and bibliographical references appear good
However, the empirical part should be better defined
The model should be better explained and above all the results should be better highlighted.
The conclusions should be strengthened by highlighting the relevance of the results obtained in relation to previous studies on the subject.

Author Response

Thank you for valuable comments

Round 2

Reviewer 1 Report

The manuscript is improved in the right direction. However, some of the important recommendations are not covered.

Author Response

Thank you for the comments! 

Each of them will be addressed separately.

Reviewer 2 Report

I am satisfied with the changes made.

Author Response

No comments

Round 3

Reviewer 1 Report

General comments

The manuscript is well improved, but still has some major opportunities for improvement.

Also, the reporting of results or discussion does not explicitly answer the research questions. The discussion section is very general and does not give a clear picture of the value of the research. The findings are not referenced to previous research.

Introduction

?? The research aims to answer the study: ...

Empirical dataset and framework

Reporting does not follow the logic of the research questions.

Recommendation: structure the report according to the research question.

Discussion

Discussion is general, brief and narrow. Incomplete explanation of the contribution (practical and scientific). Lack of references to other research findings.

Recommendations:

Recommendations: Follow best practices for scientific writing. Base the section on the interpretation of the specific findings of the research. A scientific conclusion must refer to other research findings and place the presented study within a particular research setting. Explain clearly the outcome of the research in light of scientific development and support the claims with appropriate previous research.

Author Response

Thank you for the comments. The author has responded to each comment separately.
